# The Phosphinate Group in the Formation of 2D Coordination Polymer with Sm(III) Nodes: X-ray Structural, Electrochemical and Mössbauer Study

**DOI:** 10.3390/ijms232415569

**Published:** 2022-12-08

**Authors:** Ruslan P. Shekurov, Mikhail N. Khrizanforov, Almaz A. Zagidullin, Almaz L. Zinnatullin, Kirill V. Kholin, Kamil A. Ivshin, Tatiana P. Gerasimova, Aisylu R. Sirazieva, Olga N. Kataeva, Farit G. Vagizov, Vasili A. Miluykov

**Affiliations:** 1Arbuzov Institute of Organic and Physical Chemistry, FRC Kazan Scientific Center of RAS, Arbuzov Str. 8, 420088 Kazan, Russia; 2A.M. Butlerov Chemistry Institute of the Kazan Federal University, Kremlevskaya Str. 18, 420008 Kazan, Russia; 3Institute of Physics, Kazan Federal University, Kremlevskaya str. 18, 420008 Kazan, Russia; 4Department of Physics, Kazan National Research Technological University, 68 Karl Marx Street, 420015 Kazan, Russia

**Keywords:** metal–organic frameworks, coordination polymers, rare-earth metals, lanthanides, phosphinates, elecrochemisty, ferrocene derivatives, redox

## Abstract

A coordination polymer has been synthesized using ferrocene-based ligand-bearing phosphinic groups of 1,1′-ferrocene-diyl-bis(*H*-phosphinic acid)), and samarium (III). The coordination polymer’s structure was studied by both single-crystal and powder XRD, TG, IR, and Raman analyses. For the first time, the Mössbauer effect studies were performed on ferrocenyl phosphinate and the polymer based on it. Additionally, the obtained polymer was studied by the method of cyclic and differential pulse voltammetry. It is shown that it has the most positive potential known among ferrocenyl phosphinate-based coordination polymers and metal–organic frameworks. Using the values of the oxidation potential, the polymer was oxidized and the ESR method verified the oxidized Fe(III) form in the solid state. Additionally, the effect of the size of the phosphorus atom substituent of the phosphinate group on the dimension of the resulting coordination compounds is shown.

## 1. Introduction

Polynuclear complexes of rare-earth metals are of particular interest due to unusual electronic interactions leading to a variety of physicochemical properties [1,2,3]. In this regard, the development of approaches to the directed synthesis of polynuclear complexes with desired properties is an urgent scientific problem. The design of molecular magnets is one of the relatively new scientific directions associated with the synthesis of high-dimensional systems. Molecular magnets can be used in various fields: magnetic protection against low-frequency fields, transformers and generators of low weight, scientific instrumentation, cryogenic engineering, information technology, medicine, and energy. Luminescent molecular materials based on complexes of rare-earth metals are a promising research topic for scientists [4]. Possible applications are luminescent polymer films, active optical fibers for data transmission, and light-emitting devices.

The interest in coordination compounds and metal–organic frameworks (MOFs) of rare-earth elements is currently high [5]. Only a few ferrocene-based complexes of rare-earth metals are known [6,7]. At the same time, the incorporation of ferrocene units into coordination polymers (CP) has attracted much attention due to the ability of an electron donor, reversible redox chemistry, steric properties, and rapid functionalization of this stable fragment. Such compounds also represent the initial type of heterometallic complexes, since they are potentially capable of combining certain properties of the metal ion and the organometallic fragment [8,9,10].

There are two main approaches for the synthesis of ferrocene containing 2D and 3D MOFs: (1) the use of trivalent metals with a high coordination number and (2) the use of additional bi- and polydentate neutral ligands as linkers. Both methods have been successfully used to obtain coordination polymers based on ferrocenylcarboxylic acids [11,12]. 

Compared with carboxylic acids, phosphinate derivatives are characterized by a wide structural diversity, which is mainly achieved due to the ability of phosphinate groups to exhibit various structural functions [13,14,15,16,17,18,19,20]. The variety of coordination polymers and MOF obtained depends on both the substituent at the phosphorus atom of the phosphinate group and the nature of the complexing cation, as well as the experimental conditions (solvent and temperature). Thus, in the case of cobalt (II) nitrate and 1,1′-ferrocenylenbis(*H*-phosphinic) acid, a 1D helical coordination polymer is formed in a methanol-DMF solution at 80 °C [19], whereas in water at 25 °C a 2D porous coordination layered polymer is formed from the same components [14]. In this system cobalt (II) nitrate—1,1′-ferrocenylenbis(*H*-phosphinic) acid the synthesis of 3D MOFs can also be achieved by the inclusion of additional ligands. So, in the case of 4,4′-bipyridine addition, a 3D metal–organic framework is formed, where layers of 2D polymer are cross-linked by a 4,4′-bipyridyl bridging ligand, resulting in a 3D network. In all cases, only one polymer phase is formed. These possibilities are realized due to the conformational lability of phosphinate groups, which makes them very attractive for the construction of metal–organic frameworks. In this regard, it is of interest to investigate the complexing properties of 1,1′-ferrocendiyl-bis(phosphinic acids) with respect to rare-earth metal compounds. It should be noted that to date, only one work, devoted to complexes of lanthanides with phosphinic acids, is known [21]. 

Earlier in our group, only discrete molecules based on Fe (III) and 1,1′-ferrocendiyl-bis(phenylphosphinic) acid were obtained as a ligand with a chelating type of metal-ligand coordination [22,23]. The use of trivalent metals such as Al (III) and Fe (III) with 1,1′-ferrocendiyl-bis(*H*-phosphinic) acid results in the formation of porous MOFs as aerogels of unknown structure [24,25]. Therefore, the study of the complex formation of ferrocenyl phosphinates with trivalent lanthanides leaves a chance to obtain coordination polymers due to the implementation of a different type of metal-ligand interaction. Earlier, it was not possible to determine the structure and coordination motifs on the basis of a series of europium, dysprosium, and yttrium atoms [26].

Herein, we present for the first time the X-ray structural analysis of a 2D metal–organic framework based on 1,1′-ferrocendiyl-bis(*H*-phosphinic) acid and samarium (III). In addition to the chelating type of coordination, which was previously demonstrated in the discrete complex of ferrocenylphenylphosphinate, a bridging type of coordination is also realized, which allows the formation of the polymer. The aim of this work is also to study the effect of the size of the substituent at the phosphorus atom of the phosphinate group on the size of the resulting coordination compounds.

## 2. Results

We found that heating a mixture of 1,1′-ferrocenylenbis(*H*-phosphinic) acid **1** and Sm (III) chloride in water at 100 °C leads to the formation of a layered two-dimensional coordination polymer **2** (Figure 1). This coordination polymer **2** was studied by X-ray, TG, IR, and powder XRD analyses. 

The thermogravimetric analysis curve clearly demonstrates that the CP **2** is stable in the 25–300 °C range (Figure 1). At higher temperatures, phosphinate groups are oxidized (weight gain due to the addition of oxygen atoms). Then after 600 °C, the sample decomposes, reaching a stable plateau at 800 °C. Most likely, iron and samarium phosphinates and oxides remain. A similar picture of thermal decomposition was observed for CP based on Dy, Eu, and Y [26]. Metal–organic frameworks based on lanthanides don’t contain solvent (water) in the crystal lattice, which is confirmed by the lack of bending until thermal decomposition on the TG curves.

Analysis of the powder diffraction pattern data also shows that all coordination polymers based on Sm, Dy, Eu, and Y are isostructural and phase pure (Figure 2) [26].

According to X-ray single crystal diffraction compound **2** is a two-dimensional 2D CP with the ligands being in eclipsed conformations with torsion P-C-C-P angles equal to c.a. 76° and c.a. 71° and exhibiting two types of coordination. Two ligands comprise a double O-P-O bridge between two Sm-ions with the formation of eight-membered cycles. The other two ligands form mono bridges between Sm (III) ions in different directions, thus binding these eight-membered rings, forming a two-dimensional structure running along the 0b axis and diagonal of the a0c plane (Figure 3).

The Sm (III) ions have a pseudo-octahedral environment (the d(O–Sm) = 2.273(2) − 2.309(3) Å), which is inessential for complexes obtained in an aqueous medium. According to CCDC, the most common coordination numbers for Sm (III) ion in ferrocene-containing structures are 8 and 9, and only one with a coordination number equal to 6, which was synthesized in an anhydrous environment [27]. The Sm (III) ions are coordinated by six oxygen atoms from four phosphinate ligands. Planes of cyclopentadienyl rings of phosphinate ligands forming adjacent eight-membered cycles are almost perpendicular with an angle of about 86°, whereas the chelated and bridging ligands within one eight-membered cycle are parallel.

It should be noted that the type of metal coordination in the series of 1,1′-ferrocendiyl-bisphosphinates is similar (Figure 4). The equivalence of the rotation angles of the cyclopentadienyl rings of the *tris*-chelate complex and the coordination polymer is observed. In the present case, the “third” ligand obtained using the inversion center (1 − x, 1 − y, 1 − z) is displaced in the [111] and the torsion O-P-C-C angle is different and equal to 96.3° compared to 10.3° and 11.1° for ligands in an asymmetric unit. This rotation of the cyclopentadienyl ring allows it to bind to another Sm(III) ion. For 1,1′-ferrocendiyl-bisphosphinate complex all cyclopentadienyl rings lie in the same plane.

However, the smaller volume of the substituent at the phosphorus atom in 1,1′-ferrocendiyl-bis(*H*-phosphinic acid) compared with 1,1′-ferrocendiyl-bis(phenylphosphinic acid) causes the free rotation of phosphinate groups around P–C-bonds, C-C-P-H torsion angles vary from −106.2° to 148°, as a result of which there is a bridging type of coordination of 1,1′-ferrocendiyl-bis(*H*-phosphinic acid) and the formation of a coordination polymer occurs. The adjacent coordination polymers are connected by C–H···O and C–H···π interactions.

The orange color of the sample is reflected in its UV/Vis spectrum. The corresponding band appears at ∼460 nm and is associated with the Fc moiety of **2** (Figure 5).

The IR and Raman spectra of coordination polymer **2** (Figure 6) are very close to the spectra of previously published Y, Dy, and Eu metal–organic frameworks with moderate shifts of bands of (H)PO_2_^−^ moiety (up to 12 cm^−1^) due to variation of Ln atom. In the IR/Raman spectra it is featured by the band of ν_as_PO_2_^−^ at 1149/1148 cm^−1^ (1161/1157 cm^−1^ for Y, 1159/1154 cm^−1^ for Dy, and 1156/1154 cm^−1^ for Eu). Positions of bands of *P*-substituted ferrocene fragment remain practically unchanged (νCC at 1426, 1385, 1370, and 1360 cm^−1^ with breathing mode of Cp^P^ at 1190 cm^−1^ in IR spectra and νCC at 1428, 1391, 1372 cm^−1^, breathing mode of Cp^P^ at 1184 cm^−1^ in the Raman spectra).

Similar to previously published spectra of Y, Dy, and Eu coordination polymers, the IR-spectrum of coordination polymer **2** contains three bands of P-H stretching vibrations at 2392, 2338, and 2319 cm^−1^ (2408/2405/2396, 2340/2339/2338 and 2322/2322/2320 cm^−1^ for Y/Dy/Eu). A similar trend is observed in the Raman spectra. It was suggested previously [26] that multiple νP-H bands are caused by the coexistence of several types of coordination in the coordination polymers: the highest band was interpreted as P-H stretching of (H)PO_2_^−^ moiety coordinated to two metal ions similar to Mn- and Co- cases [28,29], whereas bands at lower frequencies were assigned to mono-coordinated (H)PO_2_^−^ moiety, like in the case of *tris-*chelate Fe(III) complex [22]. Fortunately, for CP **2** it was managed to grow a crystal. Analysis of X-ray data completely confirms our previous assignment.

One of two (H)PO_2_^−^ moieties at the ferrocene fragment is coordinated to two Sm(III) ions, and the second one at the other Cp^P^ ring is coordinated to one metal atom. Two close bands in the latter case arise from the close localization of two O atoms from PO-groups of neighboring ferrocene fragments (2.39 Å) that is even shorter that in the Fe(III) complex (2.45 Å) (Figure 7, right). Hydrogen migrates between these two oxygen atoms providing two unequal P-H bonds.

In addition, for the first time, the Mössbauer effect studies were performed on ferrocenyl phosphinate **1** and the coordination polymer 2 based on it. Room temperature Mössbauer spectra of both **1** and **2** may be well-fitted with the single doublet component (Figure 8). The hyperfine parameters are listed in Table 1. The center shift values of these compounds are almost matching with each other and with the value for ferrocene. It shows that the electron density on ^57^Fe is nearly the same. However, a more pronounced effect was observed in the quadrupole splitting (QS) values. The QS for **2** is less than for both **1** and ferrocene. 

QS is determined by the electric field gradient (EFG) on iron nuclei. The main source of the EFG is electrons in the 3*d* shell. Thus, the reduction in QS for **2** should be related to the changes in the 3*d* orbitals relative population. However, the total occupation of these orbitals should not alter significantly since the central shift values are almost the same. The positive sign of EFG for ferrocene was reported in [30]. Therefore, considering known values of angular parts of 3*d* wavefunctions [31], we may suppose that the relative population in *d_x_*^2^_−*y*_^2^ and *d_xy_* orbitals decreases and in *d_z_*^2^, *d_xz_*, and *d_yz_* increases in **2** in comparison with **1** and ferrocene. 

**Table 1 ijms-23-15569-t001:** Room temperature ^57^Fe hyperfine parameters.

Sample	Center Shift, mm/s	Quadrupole Splitting, mm/s	Lamb–Mössbauer Factor
**1**	0.444 (5)	2.27 (1)	0.33
**2**	0.421 (5)	2.21 (1)	0.27
Ferrocene	0.44 [32]	2.40 [32]	0.07 [33]

In Figure 9, temperature dependencies of the Mössbauer spectral area normalized to room temperature are shown. With the decrease in temperature, the spectral area for the samples notably rises. This feature of the temperature dependence of the absorption line area is due to an increase in the recoilless fraction Lamb–Mössbauer factor, *f*_LM_). However, for **2**, a more explicit increase in the area was observed. Actually, it shows that the value of *f*_LM_ at room temperature for **2** is less than for **1**. Such deviation is connected to the differences in the vibrating properties of these samples. Temperature dependency of the Mössbauer absorption area may be processed within a frequently used Debye model of solids but with an effective value of vibrating mass [32]. Following the ref. [34], we also assumed that this mass is equal to the ferrocene unit, i.e., 187 amu. The best fit results are depicted by the solid lines in Figure 9. The derived effective Debye temperatures and values of *f*_LM_ at room temperature are 106(1) K and 0.33, 97(1) K, and 0.27 for **1** and **2**, respectively. The obtained values of *f*_LM_ for these samples are notably higher than for ferrocene (0.07 [33]). This may be explained by the difference in the bonding strength of the ferrocene unit. In pure ferrocene, molecules are bonded with each other within the weak van der Waals forces. The weak forces determine the low-frequency character of intermolecular vibrations and, consequently, low magnitudes of the Debye temperature and *f*_LM_ at room temperature. Vibrations of ferrocene units in **1** and **2** are expanded to higher frequencies. In the case of **1**, it is caused by the stronger intermolecular forces which are hydrogen bonds. For **2**, a ferrocene unit is incorporated into the polymer structure. However, our results show that the mean bonding strength of this unit in **2** is even less than in **1**. The two-dimensional character of bonding in the polymer structure and weak bonds in the third direction may reduce the average strength of ferrocene unit bonding.

Synthesized 2D coordination polymer **2** was also characterized by cyclic voltammetry (CV) and differential pulse voltammetry measurements. The electrochemical study of the redox properties of the Sm(III)-based coordination polymer was carried out using a carbon paste electrode based on phosphonium salt as a binder, which has proven itself for these purposes [23]. Among all known coordination polymers based on *H*-phosphinates, CP **2** has the most positive potential for ferrocene oxidation. In the series of lanthanoids, the oxidation potential decreases in the order Sm-Dy-Y-Eu (Figure 10).

The ferrocene fragment of CP **2** in the solid state reversibly oxidizes at 2.34 V vs. Ag/AgCl with ΔE = 121 mV, that is, at more positive potentials comparable to pure ferrocene (0.45 V and ∆Ep = (Epa − Epc) = 60 mV). In the cathode region, no characteristic waves are observed (up to −2.9 V). The reversibility of the process for CP **2** can be clearly seen on the semi-differential CV curve (Figure 11), where the areas of the direct and reverse currents coincide.

Iron (III) *tris*-chelate, previously described in the literature, also shifts the oxidation potential of ferrocene fragments to the positive region by 0.7 V relative to ferrocene. However, the group of lanthanides significantly complicates the oxidation of ferrocene fragments, whereas samarium is similar in properties to dysprosium, and yttrium is similar in electrochemical properties to europium. 

Using the values of oxidation potential, polymer **2** was oxidized in a phosphonium gel on a glassy carbon substrate. After passing 1F per mole of polymer **2**, the powder was examined by the ESR method, a single broad line with g = 2.09 and a width of ΔH = 600 G is recorded (Figure 12). In addition, a low-field component is observed, the position of which significantly depends on the orientation of the sample. The position of the high-field component weakly depends on the orientation of the ampoule with the sample. The figure shows the averaged spectrum of the oxidized sample after a series of ampoule rotations around its own axis. The value of the g-factor of the low-field component after averaging was 3.73. The presence of low-field components is characteristic of a high-spin iron atom. In addition, spectra with parameters close to ours are characteristic of a number of cases with Fe(III) 3d^5^ high-spin centers [35,36] that allow us to conclude the oxidation of Fe(II) into the Fe(III).

## 3. Materials and Methods

### 3.1. General

The thermogravimetric analysis (TGA) was carried out on air using a Netzsch STA 409 PC Luxx thermal analyzer and heating rate of 5°/min in the temperature region from 20 up to 1200 °C. IR spectra of solid compounds have been registered using a Bruker Vector-27 FTIR spectrometer in the 400–4000 cm^−1^ range (optical resolution 4 cm^−1^); the samples were prepared as nujol mulls. Powder X-ray diffraction data were collected on an STOE STADI P diffractometer with Cu-Kα_1_ radiation (λ = 1.5405 Å). 

### 3.2. X-ray Diffraction Analysis

Dataset for single crystal of coordination polymer **2** was collected on a Bruker Kappa APEX diffractometer with graphite-monochromated Mo Kα radiation (λ = 0.71073 Å) at 296(3) K. Programs used: data collection APEX [37], data reduction SAINT [38], multi-scan absorption correction SADABS [39], structure solution SHELXT, structure refinement by full-matrix least-squares against F2 using SHELXL [40]. Hydrogen atoms at phosphorus atoms and oxygen atoms were revealed from difference Fourier maps and refined isotropically.

Crystal data: formula C_20_H_21_Fe_2_O_8_P_4_Sm, M = 775.30 g/mol, monoclinic, space group *P*2_1_/*n* (No. 14), Z = 4, a = 15.5690(19) Å, b = 10.1925(15) Å, c = 16.5834(19) Å, β = 111.714(6)°, V = 2444.8(6) Å^3^, ρ_calc_ = 2.106 g·cm^−3^, μ = 3.849 mm^−1^, 40613 reflections collected (−20 ≤ *h* ≤ 21, −13 ≤ *k* ≤ 9, −22 ≤ *l* ≤ 21), θ range = 1.534° to 28.689°, 6136 independent (R_int_ = 0.0717) and 4855 observed reflections [I ≥ 2σ(I)], 336 refined parameters, R(F) = 0.0319, wR(F^2^) = 0.0682, max (min) residual electron density 0.632 (−1.141) e Å^−3^. CCDC 2087339 contains the supplementary crystallographic data for this paper.

### 3.3. Mössbauer Effect Studies

The transmission spectra were collected on a conventional WissEl spectrometer working in a constant acceleration mode. ^57^Co(Rh) source (from Ritverc) with an activity of about 25 mCi was used. Measurements were carried out in the temperature range of 80–295 K using CFICEV (ICE Oxford) flow cryostat equipped with CryoCon 32B controller. The velocity scale of the spectrometer was calibrated using the spectrum of thin α-Fe foil. The experimental spectra were processed using SpectrRelax 2.1 software (Moscow State University, Russia) [41]. The center shift values are given relative gravity center of α-Fe spectrum at 295 K.

### 3.4. Electrochemical Measurements

Electrochemical measurements were performed using a BASiEpsilon EClipse electrochemical analyzer (West Lafayette, IN, USA) in conjunction with an Epsilon-EC-USB-V200 potentiostat. A conventional three-electrode system was used with an RTIL-CPE as working electrodes (ð = 3 mm), an Ag/AgCl (3 M NaCl) electrode as the reference electrode, and a Pt wire as counter electrode. A 0.1 M Bu_4_NBF_4_ was used as supporting electrolyte for the measurement of current-voltage curves. The reference electrode was connected with the cell solution by a modified Luggin capillary filled with the supporting electrolyte solution (0.1 M Bu_4_NBF_4_ in CH_3_CN). Thus, the reference electrode assembly had two compartments, each terminated with an ultra-fine glass frit to separate the AgCl from the analyte.

### 3.5. ESR Measurements

ESR measurements were carried out on an X-band ELEXSYS E500 ESR spectrometer. Samples in quartz ampoules 5 mm in diameter were inserted into the ER 4102ST cavity, after which the spectrometer was tuned and the ESR spectra were recorded. Bruker E 035M teslameter was used to accurately g-factor determine.

### 3.6. Synthesis

1,1′-ferrocenylenbis(*H*-phosphinic) acid (H_2_fcd*H*p) was prepared according to a literature procedure [42]. All other chemicals and solvents were purchased reagent-grade and used as received.

Synthesis of poly(1,1′-ferrocenediyl-bis(H-phosphinate) Sm(III)) (**2**) SmCl_3_ 6H_2_O (11 mg; 0.03 mmol) and Fc(P(O)(H)OH)_2_ (19 mg; 0.06 mmol) were dissolved in 10 mL of water. After 12 h at 100 °C degrees temperature, orange crystalline precipitate of **2** was obtained. Yield: 19 mg (81.7%) based on 1,1′-ferrocenediyl-bis(*H*-phosphinic acid). Anal. Calcd. for coordination polymer **2** C_20_H_21_Fe_2_SmO_8_P_4_ (775.30 g/mol): C: 30.95; H: 2.71%. Found: C: 31.10; H: 2.77%. IR (nujol, cm^−1^): ∼3400 νOH; 3081, 3095, 3107 νCH; 2392, 2338, 2319 νPH; 1426, 1385, 1370, 1360 Cp^P^ νCC; 1190 Cp^P^ breath; 1171 Cp^P^: in-plane δCH; 1161 νasPO_2_^−^; 1068, 1030, 982 mixed vibrations: Cp^P^ δCH, δPH, ν_s_PO_2_^−^; 894, 871 Cp^P^ in-plane bendings; 844, 825 Cp^P^ γCH; 640 Cp^P^ in-plane deformations; 473 νFe-C, ring tilt.

## 4. Conclusions

A new coordination of polymer **2** using ferrocene-based 1,1′-ferrocenylenbis(H-phosphinic) acid **1** and Sm (III) chloride in water was synthesized at 100 °C. The structure and properties of coordination polymer **2** were studied by X-ray and powder XRD analyses, TG, IR, and Raman analyses. In addition, Mössbauer effect studies were performed on ferrocenyl phosphinate **1** and the coordination polymer **2** for the first time. It shows that the electron density on ^57^Fe is nearly the same; the more pronounced effect was observed in the quadrupole splitting values. Additionally, the obtained polymer was studied by the method of cyclic and differential pulse voltammetry. It is shown that CP **2** has the most positive potential known among ferrocenylphosphinate-based coordination polymers and MOFs. Using the values of the oxidation potential, polymer **2** (Fe(II)) was oxidized in a phosphonium gel on a glassy carbon substrate and the ESR method also verified the oxidized form Fe(III) in the solid state.

Additionally, the effect of the substituent size at the phosphorus atom of the phosphinate group on the dimension of the resulting CPs is shown. In the case of trivalent metals, for example, samarium (III), due to the smaller size of the hydrogen atom, the possibility of free rotation of the phosphinate group is realized, and as a consequence, the implementation of bridging by the ligand of two cations. The free rotation of *H*-phosphinate groups leads to the formation of a 2D CP with a sandwich structure. At the same time, the larger size of the phenyl group does not allow phosphinate groups to unfold freely, as a result of which only the chelating type of cation binding is realized, and as a consequence a discrete complex is formed.

## Data Availability

The data presented in this study are contained within the article or are available upon request from the corresponding author M.N.K.

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
