# Peer review of "The Phosphinate Group in the Formation of 2D Coordination Polymer with Sm(III) Nodes: X-ray Structural, Electrochemical and Mössbauer Study"

_ijms, 2022, doi:10.3390/ijms232415569_

Round 1

Reviewer 1 Report

The manuscript reports about the substituent size role at the ferrocene-based ligand on the formation of 2D Metal-Organic Framework. The research is more like a routine work on synthesis of a metal-organic framework using ferrocene ligands. I recommend this paper publishing on INTERNATIONAL JOURNAL OF MOLECULAR SCIENCES after major revision. In this reviewer's opinion, the authors need to address the following issues as much as they can, and re-submit their work.

1.   In the author's title, the highlights of the author's research is the substituent size role on the formation of Metal-Organic Framework. However, there are too few relevant studies and the authors should analyse and compare the results of multiple groups of studies to acquire systematic conclusions.

2.   There are too little basic characterisation of the MOF, such as solvent stability, UV-Vis absorption spectra, Raman spectra and so on.

3.   In the cyclic voltammetry (CV) measurements, there are no complete redox couples and multiple scans in the spectrum. However, the authors concluded the ferrocene fragment of MOF 2 reversibly oxidized in the solid state, please give a clear explanation for readers.

4.   The images can be further modified to make the reading more fluent.

Author Response

Dear Reviewer,

Herewith we resubmit a manuscript by Ruslan Shekurov, Mikhail Khrizanforov*, Almaz Zagidullin, Almaz Zinnatullin, Kirill Kholin, Kamil Ivshin, Tatiana Gerasimova, Aisylu Gilfanova, Olga Kataeva, Farit Vagizov, Vasili Miluykov «The Phosphinate Group in the Formation of 2D Coordination polymer with Sm(III) nodes: X-ray Structural, Electrochemical and Mössbauer Study» to the International Journal of Molecular Sciences for participation in the thematic issue «Synthesis and Molecular Applications of Metal-Organic Frameworks (MOFs)»

First of all, we would like to thank the Reviewers for inspective reading and reviewing of our manuscript and their valuable remarks.

We are confident that our results provide new important insights into the organic, organophosphorus and organometallic chemistry and such would be of a substantial interest to the broad interdisciplinary readership of the “International Journal of Molecular Sciences”. We therefore strongly hope that our work can be accepted for publication in your Journal.

We also provide the detailed answers to the reviewers’ remarks.

Please, do not hesitate to contact me if you have any problems or questions regarding our manuscript or if you have difficulty opening the files.

Yours sincerely,

Dr. Mikhail Khrizanforov,

Response to Reviewer 1 Comments

Q1 The article by Khrizanforov et al. reports the X-ray single-crystal structure of a Sm(III)-Ferrocene 2D coordination polymer, with an unusual six coordination mode of the Sm(III) ions. XRPD measurements confirmed the purity of the samples prepared with a series of Ln(III) ions. All characterizations have been clearly discussed. The literature has been correctly addressed (despite few examples of useless self-citations, 15-18. Reference 19 should be mentioned in the text if it does correspond to the example of 3D MOF with 4,4’-bipyridine).

Reply: Thank you very much for your comments. We have removed reference 16, but we kindly ask you to leave the rest, since we show that the resulting polymer has the most positive among all known ferrocenyl phosphinates.

Q2 The overall quality of the paper is convincing. However, I must disagree about the use of “Metal-Organic Framework” terminology here, which must be (although not strictly respected) reserved to 3D or even 2D porous coordination polymer (I recommend the lecture of this reference “James, S.L. Chem. Soc. Rev. 2003, 32, 276–288” or more recently “Oggianu, M.; Manna, F.; Ashoka Sahadevan, S.; Avarvari, N.; Abhervé, A.; Mercuri, M.L. Crystals 2022, 12, 763”). According to the crystal structure and TGA analysis, no crystallization solvent was observed, and there is no discussion about the porosity (accessible voids, use of the SQUEEZE program). Please discuss this aspect to confirm the porous nature of the coordination polymer, or rather use the term of 2D coordination polymer.

Reply: Thank you for your comment. We changed the term to coordination polymer

Q3 I don’t see how does the layered structure grow. From Figure 3 I can observe a 1D chain, so my question is how these chains extend to 2D layers? The authors should add the crystallographic axes to the structure figures.

Reply: Thank you for your comment. We have presented an image where you can see how 2D layers are formed. We also added the corresponding description to the text (highlighted in color)

Q4 Please clarify if the synthesis was performed at 100°C (2. results) or 90°C (3. materials and methods). If these corrections are considered, I would recommend the publication of the manuscript in IJMS.

Reply: Thanks for finding the typo. The synthesis was performed at 100°C

Reviewer 2 Report

The article by Khrizanforov et al. reports the X-ray single-crystal structure of a Sm(III)-Ferrocene 2D coordination polymer, with an unusual six coordination mode of the Sm(III) ions. XRPD measurements confirmed the purity of the samples prepared with a series of Ln(III) ions. All characterizations have been clearly discussed. The literature has been correctly addressed (despite few examples of useless self-citations, 15-18. Reference 19 should be mentioned in the text if it does correspond to the example of 3D MOF with 4,4’-bipyridine).

The overall quality of the paper is convincing. However, I must disagree about the use of “Metal-Organic Framework” terminology here, which must be (although not strictly respected) reserved to 3D or even 2D porous coordination polymer (I recommend the lecture of this reference “James, S.L. Chem. Soc. Rev. 2003, 32, 276–288” or more recently “Oggianu, M.; Manna, F.; Ashoka Sahadevan, S.; Avarvari, N.; Abhervé, A.; Mercuri, M.L. Crystals 202212, 763”). According to the crystal structure and TGA analysis, no crystallization solvent was observed, and there is no discussion about the porosity (accessible voids, use of the SQUEEZE program). Please discuss this aspect to confirm the porous nature of the coordination polymer, or rather use the term of 2D coordination polymer.

I don’t see how does the layered structure grow. From Figure 3 I can observe a 1D chain, so my question is how these chains extend to 2D layers? The authors should add the crystallographic axes to the structure figures.

Please clarify if the synthesis was performed at 100°C (2. results) or 90°C (3. materials and methods).

If these corrections are considered, I would recommend the publication of the manuscript in IJMS.

Author Response

Dear Reviewer,

Herewith we resubmit a manuscript by Ruslan Shekurov, Mikhail Khrizanforov*, Almaz Zagidullin, Almaz Zinnatullin, Kirill Kholin, Kamil Ivshin, Tatiana Gerasimova, Aisylu Gilfanova, Olga Kataeva, Farit Vagizov, Vasili Miluykov «The Phosphinate Group in the Formation of 2D Coordination polymer with Sm(III) nodes: X-ray Structural, Electrochemical and Mössbauer Study» to the International Journal of Molecular Sciences for participation in the thematic issue «Synthesis and Molecular Applications of Metal-Organic Frameworks (MOFs)»

First of all, we would like to thank the Reviewers for inspective reading and reviewing of our manuscript and their valuable remarks.

We are confident that our results provide new important insights into the organic, organophosphorus and organometallic chemistry and such would be of a substantial interest to the broad interdisciplinary readership of the “International Journal of Molecular Sciences”. We therefore strongly hope that our work can be accepted for publication in your Journal.

We also provide the detailed answers to the reviewers’ remarks.

Please, do not hesitate to contact me if you have any problems or questions regarding our manuscript or if you have difficulty opening the files.

Yours sincerely,

Dr. Mikhail Khrizanforov,

  Response to Reviewer 2 Comments

The manuscript reports about the substituent size role at the ferrocene-based ligand on the formation of 2D Metal-Organic Framework. The research is more like a routine work on synthesis of a metal-organic framework using ferrocene ligands. I recommend this paper publishing on INTERNATIONAL JOURNAL OF MOLECULAR SCIENCES after major revision. In this reviewer's opinion, the authors need to address the following issues as much as they can, and re-submit their work.

 Q1 In the author's title, the highlights of the author's research is the substituent size role on the formation of Metal-Organic Framework. However, there are too few relevant studies and the authors should analyse and compare the results of multiple groups of studies to acquire systematic conclusions.

Reply: Thank you very much for your comment. Your comments and questions have prompted us to substantially revised the manuscript. Also, we have changed the title.

Q2 There are too little basic characterisation of the MOF, such as solvent stability, UV-Vis absorption spectra, Raman spectra and so on.

Reply: Thank you very much for your comment. The coordination polymer’s structure was studied by both single-crystal and powder XRD, TG, IR, Raman analyses. For the first time, the Mössbauer effect studies were performed on ferrocenyl phosphinate and the polymer based on it. Additionally, the obtained polymer was studied by the method of cyclic and differential pulse voltammetry. It is shown that it has the most positive potential known among ferrocenyl phosphinate based coordination polymers and MOFs. Using the values of the oxidation potential, the polymer was oxidized and ESR method verified the oxidized Fe(III) form in the solid state.

Q3  In the cyclic voltammetry (CV) measurements, there are no complete redox couples and multiple scans in the spectrum. However, the authors concluded the ferrocene fragment of MOF 2 reversibly oxidized in the solid state, please give a clear explanation for readers.

Reply: The reversibility of the process for CP 2 can be clearly seen on the semi-differential CV curve, where the areas of the direct and reverse currents coincide. Relevant text with the figure has been added to the text.

Q4.   The images can be further modified to make the reading more fluent.

Reply: Thank you for your comment. It's been done.

Round 2

Reviewer 1 Report

My issues have been addressed. I will accept

Author Response

Dear Reviewer,

We would like to thank you for inspective reading and reviewing of our manuscript and their valuable remarks. 

On behalf of the authors,
Yours sincerely,

Dr. Mikhail Khrizanforov, 
A.E.Arbuzov Institute of organic and physical chemistry Russian academy of sciences, Arbuzov Str., 8, 420088 Kazan, Russia,